# SUPER-RESOLUTION VIA CONDITIONAL IMPLICIT MAXIMUM LIKELIHOOD ESTIMATION

## ABSTRACT

Single-image super-resolution (SISR) is a canonical problem with diverse applications. Leading methods like SRGAN (Ledig et al., 2017) produce images that contain various artifacts, such as high-frequency noise, hallucinated colours and shape distortions, which adversely affect the realism of the result. In this paper, we propose an alternative approach based on an extension of the method of Implicit Maximum Likelihood Estimation (IMLE) (Li & Malik, 2018). We demonstrate greater effectiveness at noise reduction and preservation of the original colours and shapes, yielding more realistic super-resolved images.

## 1 INTRODUCTION

The problem of single-image super-resolution (SISR) aims to output a plausible high-resolution image that is consistent with a given low-resolution image. The key challenge arises from the fact that the problem is ill-posed – given the same low-resolution image, there are many different high-resolution images that would be the same as the low-resolution image upon downsampling. This problem becomes more severe as the upscaling factor increases, since there are more possible ways to fill in missing pixels as the number of such pixels grows. Therefore, to solve this problem, all methods impose a prior over images either explicitly or implicitly, so that among all possible high-resolution images that are consistent with the low-resolution image, some are preferred over others. The literature on SISR is vast; see Yang et al. (2014) and Nasrollahi & Moeslund (2014) for comprehensive overviews.

Classical approaches, such as bilinear, bicubic or Lanczos filtering (Duchon, 1979), leverage the insight that neighbouring pixels usually have similar colours and generate a high-resolution output by interpolating between the colours of neighbouring pixels according to a predefined formula. The drawback of these approaches is that they lead to an over-smoothed outputs without clear edges. There are some edge-based methods that aim to address this issue e.g.: (Li & Orchard, 2001).

Data-driven approaches, on the other hand, make use of training data to assist with generating high-resolution predictions. Exemplar-based approaches, e.g.: (Freeman et al., 2002; Freedman & Fattal, 2011; Glasner et al., 2009), directly copy pixels from the most similar patches in the training dataset. In contrast, optimization-based approaches, e.g.: (Elad & Aharon, 2006; Huang et al., 2015; Gu et al., 2015), formulate the problem as a sparse recovery problem, where the dictionary is learned from data. Some methods, e.g.: (Yang et al., 2010), combine the two approaches and use a dictionary of image patches. Some drawbacks are that the resulting images often have artifacts due to patches not blending well together and that generating a prediction is quite computationally expensive.

One way around the latter problem is to treat the problem simply as a regression problem e.g.: (Dong et al., 2016), where the input is the low-resolution image, and the target output is the high-resolution image. A highly expressive regression model, like a deep neural net, is trained on the input and target output pairs to minimize some distance metric between the prediction and the ground truth. While this indeed makes prediction faster, it effectively assumes a single plausible output for each input and ignores the multi-modality that is inherent in the problem. This comes at a cost: because the model is only allowed to generate a single image even though multiple images are plausible, it has to hedge its bets. If the loss is Euclidean, then the model would predict the mean of the plausible images in order to minimize the loss.

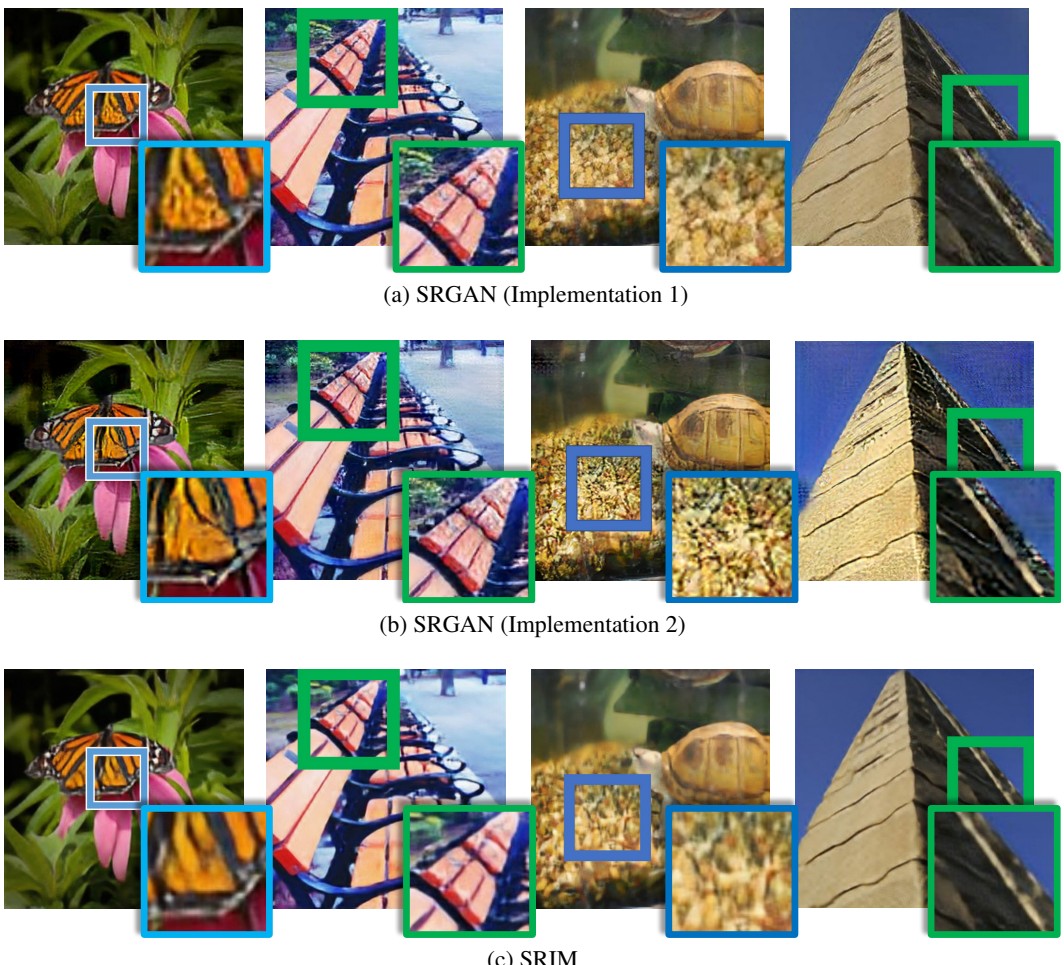

(a) SRGAN (Implementation 1)

(b) SRGAN (Implementation 2)

(c) SRIM

Figure 1: Results of top-ranked public implementations of SRGAN and the proposed method (SRIM). Notice the artifacts in the results of SRGAN implementations in column (1) at the centre of the wing, which has high-frequency noise (spots) and distorted lines (2) on back of the bench, which has high-frequency noise and hallucinated colours (3) on the gravel, which has noise that results in a grainy appearance in the result of Implementation 1, and excess contrast in the result of Implementation 2, and (4) along the edge of the pyramid, which has noise in the background in the results of both implementations and hallucinated colours in the result of Implementation 2.

Instead of forcing the model to produce a point estimate, a better alternative would be to model the full conditional distribution, i.e.: the distribution of high-resolution images given low-resolution images. This calls for the use of a probabilistic model to estimate it. One option is to use a prescribed probabilistic model, which constrains the distribution to lie in the exponential family, and learn the sufficient statistics (Bruna et al., 2015). Another option is to use an implicit probabilistic model, which models the output as a parameterized deterministic transformation of a standard Gaussian random variable and the variable the distribution is conditioned on, which in this case is the low-resolution input image. More concretely, given an input $\mathbf{x}$, the following procedure can be used to sample the output $\tilde{\mathbf{y}}$ from the model:

1. Sample $\mathbf{z} \sim \mathcal{N}(0, \mathbf{I})$
2. Return $\tilde{\mathbf{y}} := T_\theta(\mathbf{x}, \mathbf{z})$

Implicit models offer more flexibility than prescribed models, since $T_\theta$ can be any arbitrary function, including a neural net. Training such a model, however, is non-trivial, because computing the gradient of the log-likelihood of implicit models is in general difficult, and so it not clear how to maximize

likelihood. A popular alternative introduced by generative adversarial nets (GANs) (Goodfellow et al., 2014; Gutmann et al., 2014) is to minimize the distinguishability between generated samples and real data. Unfortunately, GANs are not able to maximize likelihood (Goodfellow, 2014) and suffers from a number of issues, the most notable being mode collapse/dropping (Goodfellow et al., 2014; Arora & Zhang, 2017) and training instability (Goodfellow et al., 2014; Arora et al., 2017).

The leading approach for super-resolution is based on GANs, known as SRGAN (Ledig et al., 2017). There are several public implementations available; we took the two highest ranked implementations on Github [1] and generated results using the provided pre-trained models, which are shown in Figure 1. (Later in the experiments section, we perform a comparison to a SRGAN model trained on the same data that we used to train our method.) Upon closer examination, the results contain some unrealistic artifacts, including instances of high-frequency noise, hallucinated colours and shape distortions. This may be a result of the discriminator having finite capacity and so may not be able to tell apart a real image and a synthetic image with artifacts like high frequency noise. This is consistent with empirical findings in the adversarial example literature (Szegedy et al., 2013), which observed that deep neural nets can be easily fooled by examples with adversarially chosen high-frequency noise.

In this paper, we explore an alternative approach, which extends the recently proposed method of Implicit Maximum Likelihood Estimation (IMLE) (Li & Malik, 2018) for the problem of super-resolution. Like GANs, IMLE is a likelihood-free method for training implicit models; unlike GANs, it is equivalent to maximum likelihood (under some conditions). It has a number of advantages: it can preserve all modes, make use of all available training data and train reliably. In this paper, we develop a number of improvements to IMLE motivated by the requirements of the super-resolution problem. These include (1) a new formulation for modelling conditional distributions, (2) a new loss that captures perceptual similarity, (3) a two-stage architecture that super-resolves to higher resolutions incrementally and (4) a new hierarchical sampling procedure that improves the sample efficiency. We will refer to the proposed method as Super-Resolution Implicit Model, or SRIM for short.

We demonstrate that SRIM is able to produce super-resolved images that are less noisy and more faithfully preserve the original colours and shapes in the input image compared to SRGAN. Also, SRIM demonstrated the capability of learning different modes of the conditional distribution and is able to generate multiple plausible hypotheses in regions where the input image is ambiguous. Moreover, we show that our method outperforms SRGAN in terms of realism, as perceived by human evaluators, as well as on two standard quantitative metrics for this task, PSNR and SSIM. The results of the proposed method are shown in Figure 1 in the bottom row.

## 2 METHOD

### 2.1 BACKGROUND

Given a low-resolution image, the problem of super-resolution problem requires the prediction of a plausible version of the image at a higher resolution. More formally, given a low-resolution image $\mathbf{x} \in [0, 255]^{h \times w \times 3}$, the goal is to predict a plausible high-resolution image $\tilde{\mathbf{y}} \in [0, 255]^{H \times W \times 3}$ that when downsampled, is consistent with $\mathbf{x}$, where $M > m$ and $N > n$.

This problem is ill-posed, and there could be many different plausible high-resolution images that are all consistent with the original low-resolution image. In other words, the conditional distribution $p(\tilde{\mathbf{y}}|\mathbf{x})$ is typically multimodal. Therefore, it is important to model the full distribution $p(\tilde{\mathbf{y}}|\mathbf{x})$, rather than just a point estimate.

We model $p(\tilde{\mathbf{y}}|\mathbf{x})$ using a probabilistic model parameterized by parameters $\theta$, hereafter denoted as $p(\tilde{\mathbf{y}}|\mathbf{x}; \theta)$. To learn $\theta$, we would like to maximize the log-likelihood of the ground truth outputs (high-resolution images) given the corresponding inputs (low-resolution images). That is, given a training dataset $\mathcal{D} = \{(\mathbf{x}_1, \mathbf{y}_1), \ldots, (\mathbf{x}_n, \mathbf{y}_n)\}$, we would like to solve the following optimization problem:

---

[1] https://github.com/tensorlayer/srgan (Implementation 1) and https://github.com/brade31919/SRGAN-tensorflow (Implementation 2)

$$\hat{\theta}_{\text{MLE}} := \arg\max_{\theta} \sum_{i=1}^{n} \log p(\mathbf{y}_i | \mathbf{x}_i; \theta)$$

We choose $p(\tilde{\mathbf{y}}|\mathbf{x}; \theta)$ to be an implicit probabilistic model, that is, a probabilistic model whose density cannot necessarily be expressed explicitly, but can be represented more easily in terms of a sampling procedure. The model we use is defined by the following sampling procedure:

1. Sample $\mathbf{z} \sim \mathcal{N}(0, \mathbf{I})$
2. Return $\tilde{\mathbf{y}} := T_{\theta}(\mathbf{x}, \mathbf{z})$

Here, $T_{\theta}$ is a deep convolutional neural net that takes in two inputs, the input $\mathbf{x}$ and the latent noise vector $\mathbf{z}$. Such a model has the advantage of being high expressive, but introduces difficulties for training, because its density and therefore the log-likelihood cannot in general be computed tractably. This necessitates the use of a likelihood-free parameter estimation method.

In this paper, we leverage a recently introduced method of Implicit Maximum Likelihood Estimation (Li & Malik, 2018), which is able to maximize likelihood (under appropriate conditions) without needing to compute it, in the unconditional setting. More concretely, given a set of examples $\mathbf{y}_1, \ldots, \mathbf{y}_n$, and an implicit probabilistic model $p(\tilde{\mathbf{y}}; \theta)$, IMLE generates i.i.d. samples $\tilde{\mathbf{y}}_1^{\theta}, \ldots, \tilde{\mathbf{y}}_m^{\theta}$ from $p(\tilde{\mathbf{y}}; \theta)$ and tries to find a parameter setting that where each data example is close to its nearest sample in expectation. More precisely, IMLE solves the following optimization problem:

$$\hat{\theta}_{\text{IMLE}} := \arg\min_{\theta} \mathbb{E}_{\tilde{\mathbf{y}}_1^{\theta}, \ldots, \tilde{\mathbf{y}}_m^{\theta}} \left[ \sum_{i=1}^{n} \min_{j \in \{1, \ldots, m\}} \left\| \tilde{\mathbf{y}}_j^{\theta} - \mathbf{y}_i \right\|_2^2 \right]$$

Note that computing the minimum inside the expectation is a nearest neighbour search problem. While this is commonly believed to be computationally expensive in high dimensions, thanks to recent advances in fast nearest neighbour search algorithms (Li & Malik, 2016; 2017), this is no longer an issue. In fact, backpropagation dominates the execution time, and so nearest neighbour search takes less than 1% of the execution time of the entire algorithm.

## 2.2 Conditional IMLE for Super-resolution

To adapt IMLE for the super-resolution setting, we extend IMLE in the following ways:

- Since the original IMLE is designed to model the marginal distribution $p(\tilde{\mathbf{y}})$, we modify the objective to model a family of conditional distributions, i.e.: $\left\{ p(\tilde{\mathbf{y}}|\mathbf{x}) \,|\, \mathbf{x} \in [0, 255]^{h \times w \times 3} \right\}$.
- Instead of defining a loss directly on the output, we use a loss on features of the output to capture perceptual similarity.
- We propose a two-stage architecture, where each stage upscales the input image by a factor of two (the width and height are each doubled, resulting in a quadrupling in the number of pixels).
- We propose a hierarchical sampling scheme, where only parts of the random noise vectors are sampled at a time, which then influence the pool of samples that are generated later on.

The modified algorithm is presented in Algorithm 1.

### 2.2.1 Conditional IMLE

The probabilistic model $p(\tilde{\mathbf{y}}|\mathbf{x}; \theta)$ now models a different distribution for every value of $\mathbf{x}$, which requires two changes to be made to the IMLE algorithm. First, the value of $\mathbf{x}$ must be provided to the transformation function $T_{\theta}$ in order to sample from the correct distribution. Second, the samples for different values of $\mathbf{x}$ must be kept separate since they are from different distributions. Consequently, for each input data example $\mathbf{x}_i$, we must look for the nearest sample in among the samples generated from $p(\tilde{\mathbf{y}}|\mathbf{x}_i)$.

### 2.2.2 FEATURE SPACE

Instead of defining the loss on the outputs directly, we map both the samples and the ground truth images to a feature space and apply the loss on the features. More precisely, the objective function becomes the following, where $\phi$ is a fixed predefined function:

$$\hat{\theta}_{\text{IMLE}} \coloneqq \arg\min_{\theta} \mathbb{E}_{\{\tilde{\mathbf{y}}_{i,j}^{\theta}\}_{i\in[n],j\in[m]}} \left[ \sum_{i=1}^{n} \min_{j\in\{1,\dots,m\}} \left\| \phi(\tilde{\mathbf{y}}_{i,j}^{\theta}) - \phi(\mathbf{y}_i) \right\|_2^2 \right]$$

In our context, $\phi$ maps a high-resolution image to the concatenation of the original pixels and the appropriately scaled activations of some layers in the VGG-19 convolutional neural net when evaluated on the image. More formally, $\phi(\mathbf{y}) \coloneqq (\ \alpha_1\phi_1(\mathbf{y}) \quad \cdots \quad \alpha_c\phi_c(\mathbf{y})\ )^{\top}$. We chose $c = 3$, $\phi_1(\mathbf{y})$ to be the identity function (so it outputs the raw pixel values), $\phi_2(\mathbf{y})$ to be the pre-activations of the conv2_2 layer and $\phi_3(\mathbf{y})$ to be the pre-activations of the conv4_4 layer. We choose the weights $\alpha_k$'s so that the means of the different feature components are of the same order of magnitude.

---

**Algorithm 1** Conditional Implicit Maximum Likelihood Estimation (IMLE) Procedure

---

**Require:** The set of low-resolution images $X = \{\mathbf{x}_i\}_{i=1}^{n}$, the set of corresponding high-resolution
   images $Y = \{\mathbf{y}_i\}_{i=1}^{n}$ and a feature extraction function $\phi$
   Initialize the parameters $\theta$ of the model/transformation function $T_{\theta}$
   **for** $p = 1$ **to** $N$ **do**
      Pick a random batch $S \subseteq \{1, \dots, n\}$
      **for** $i \in S$ **do**
         Randomly generate i.i.d. $m$ noise vectors $\mathbf{z}_1, \dots, \mathbf{z}_m$
         $\tilde{\mathbf{y}}_{i,j} \leftarrow T_{\theta}(\mathbf{x}_i, \mathbf{z}_j)\ \forall j \in [m]$
         $\sigma(i) \leftarrow \arg\min_j \|\phi(\mathbf{y}_i) - \phi(\tilde{\mathbf{y}}_{i,j})\|_2^2\ \forall j \in [m]$
      **end for**
      **for** $q = 1$ **to** $M$ **do**
         Pick a random mini-batch $\tilde{S} \subseteq S$
         $\theta \leftarrow \theta - \eta\nabla_{\theta}\left(\frac{n}{|\tilde{S}|}\sum_{i\in\tilde{S}}\left\|\phi(\tilde{\mathbf{y}}_{i,\sigma(i)}) - \phi(\mathbf{y}_i)\right\|_2^2\right)$
      **end for**
   **end for**
   **return** $\theta$

---

### 2.3 TWO-STAGE ARCHITECTURE

We observe that the super-resolution problems can be decomposed into a sequence of easier super-resolution problems, each of which upscales by a smaller factor. To super-resolve images by a large factor, we can therefore have multiple sub-networks, each of which super-resolves by a small factor. We note that the idea of a multi-stage architecture is not new; it has also been used in other methods that adopt coarse-to-fine approaches (Denton et al., 2015; Lai et al., 2017; Wang et al., 2018b; Karras et al., 2017). In our setting, we have two sub-networks, each of which upscales by a factor of 2, and stack them on top of one another, thereby allowing the overall network to upscale by a factor of 4. The lower sub-network takes the original low-resolution image and a random noise vector as input, whereas the upper sub-network takes the output of the lower sub-network and a different random noise vector as input. This can be viewed as a decomposition of the source of randomness into two parts, one that is only used to model the uncertainty at a coarse level, and one that can be used to model the uncertainty at a fine level.

### 2.4 HIERARCHICAL SAMPLING

We observe that in the conditional IMLE algorithm, only the sample that is nearest to each data example is used in the loss; all other samples are discarded. We could therefore improve sample efficiency by only sampling the regions of the latent noise space that are most likely to result in a sample that is close to the data example. To this end, we sequentially generate the noise vectors

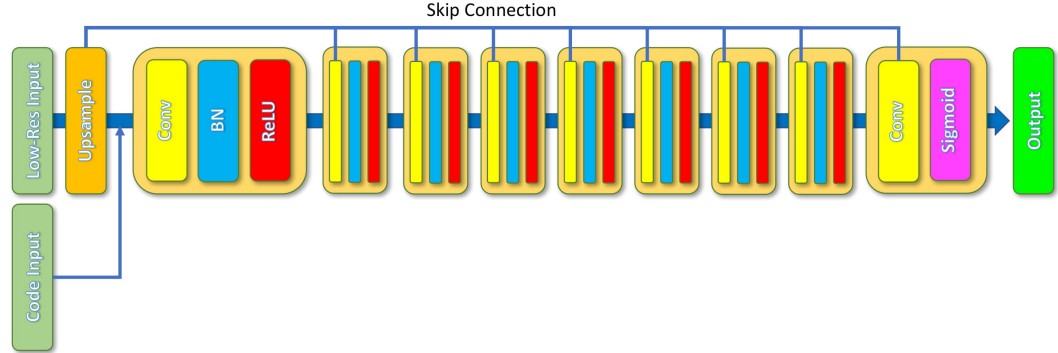

Figure 2: Architecture of one sub-network.

for the lower and upper sub-network, which we will refer to as the lower and upper noise vectors. More specifically, we first generate lower noise vectors and then compute the outputs of the lower sub-network. Then, we compare them to the ground truth downsampled to the same resolution and choose the noise vector that corresponds to the nearest sample to the downsampled ground truth. Conditioned on this lower noise vector, we then generate upper noise vectors and select the noise vector that results in a sample that is closest to the ground truth.

### 2.5 ARCHITECTURAL AND IMPLEMENTATION DETAILS

The model $T_\theta$ is a feedforward convolutional neural net with skip connections. Each sub-network consists of one bilinear upsampling layer, which upsamples the input image by a factor of 2 (doubles the width and height), followed by 9 convolution layers. Every convolution layer has a kernel size of $5 \times 5$ with 64 output channels, except for the last one, which has 3 output channels, each corresponding to a different colour channel. Between each pair of adjacent convolution layers, there is one batch normalization layer, followed ReLU activations. The last convolution layer has sigmoid activations to ensure the output is bounded between 0 and 1. The net also features skip connections from the bilinearly upsampled input image to each of the convolution layers in the sub-network except for the first one, since it is directly connected to the bilinearly upsampled input already. For the skip connections to layers in the lower sub-network, the input image is upsampled by a factor of 2, whereas for the skip connections to layers in the upper sub-network, the input image is upsampled by a factor of 4.

When computing VGG-19 activations of samples and ground truth images, we first scale up range of pixel values from $[0, 1]$ to $[0, 255]$ and then subtract then mean pixel value computed over the ImageNet dataset. Because the activations are very high-dimensional, we randomly project them to a lower dimensional space using a random Gaussian matrix on the GPU before transferring them to the CPU to keep latency and memory usage low.

## 3 EXPERIMENTS

### 3.1 DATA

For the problem of super-resolution, we can generate the training data by taking high-resolution images and downsampling them. The downsampled images will serve as input, and the original images will serve as the ground truth. The data for all experiments are from the ImageNet ILSVRC-2012 dataset, which contains one thousand categories, each with around a thousand images. We used a subset of the data to train all models, which consists of 20 categories with 150 images each, which forms a total of 3000 images. The ground truth images have a resolution of $256 \times 256$, which are obtained by anisotropic scaling of the original images. The input images have a resolution of $64 \times 64$, which are obtained by downsampling the $256 \times 256$ images. The images used for training and testing are disjoint.

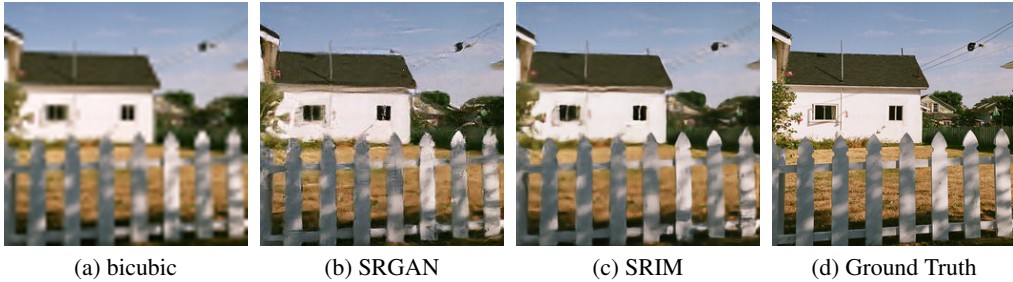

|            |            |            |                 |
| (a) bicubic | (b) SRGAN | (c) SRIM | (d) Ground Truth |

Figure 3: Representative random samples from (a) bicubic interpolation, (b) SRGAN, (c) SRIM and (d) the ground truth.

|      | Bicubic | SRGAN | SRIM | Truth |
|------|---------|-------|------|-------|
| PSNR | 23.38   | 24.06 | **25.36** | $\infty$ |
| SSIM | 0.6832  | 0.6694 | **0.7153** | 1 |

Table 1: Comparison of mean PSNR and SSIM values achieved by bicubic interpolation, SRGAN, SRIM and the ground truth, where the mean is taken over different images in the test set.

## 3.2 EVALUATION

### 3.2.1 BASELINES

We compare our results to the leading method, super-resolution generative adversarial network (SR-GAN), and the de-facto standard used most often in practice, bicubic interpolation. Both SRGAN and our method are trained on the same training data. We used the top-ranked public implementation of SRGAN on Github[2] for all experiments.

### 3.2.2 QUANTITATIVE METRICS

Two standard automated quantitative metrics used in image compression are PSNR and SSIM (Wang et al., 2004). We compute these metrics on the luminance channel for all methods using the daala package[3]. We compare our result to those of bicubic interpolation and SRGAN. All methods super-resolved inputs by a factor of 4 to generate outputs that match the ground truth resolution.

As shown in Table 1, our model (SRIM) outperforms bicubic interpolation and SRGAN in terms of both PSNR and SSIM.

### 3.2.3 QUALITATIVE ASSESSMENT

For the task of super-resolution, it is critical to preserve the fidelity to the original image, which means that the original colours, contours and shapes should be preserved and no implausible patterns should be hallucinated. On the other hand, unlike other image generation tasks, sharpness should not come at the expense of unrealistic artifacts and spurious hallucinations.

As shown in Figures 4, 5, 6, in general, our method produces fewer artifacts, maintains the real colour tone and preserves the original shapes in the input image. On the other hand, for the SRGAN results, high-frequency noise can be clearly seen on the table surface in Figure 6 and on the cloth in Figure 6, whereas our results do not contain such artifacts. Colour hallucinations are observed on the fence in Figure 5 and on Figure 6 in the SRGAN results, which are absent in SRIM result. In addition, SRGAN produces obvious distortions to the shapes of objects highlighted in yellow in Figures 5 and 6: we can see that the brow is extended and the eye has lost its original shape. On the other hand, our method was able to avoid this issue.

---

[2]https://github.com/tensorlayer/srgan
[3]https://github.com/xiph/daala

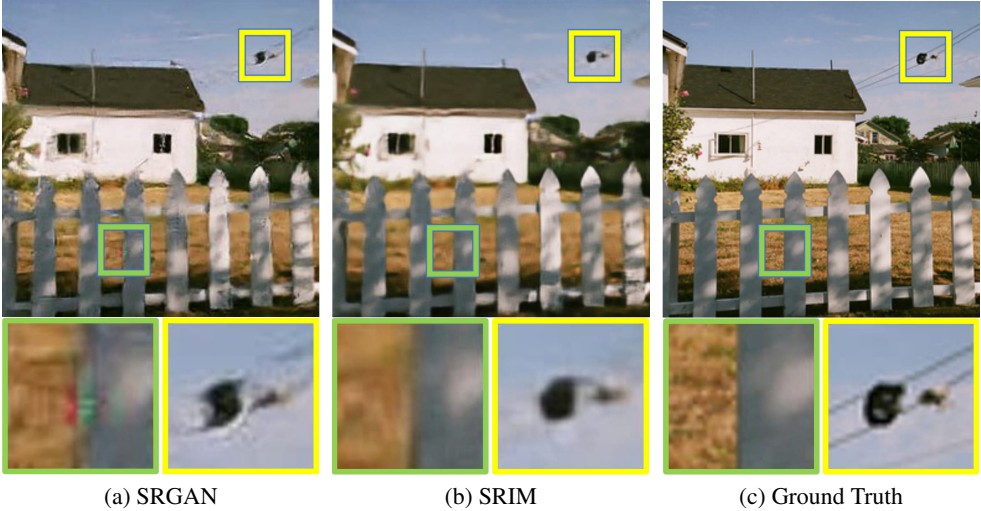

(a) SRGAN          (b) SRIM          (c) Ground Truth

Figure 4: Qualitative comparison of the results of (a) SRGAN, (b) SRIM and (c) the ground truth. Hallucinated colours and shape distortions are visible on the edge of the fence picket and near the overhead cables respectively in the output of SRGAN, whereas they are not present in the output of SRIM.

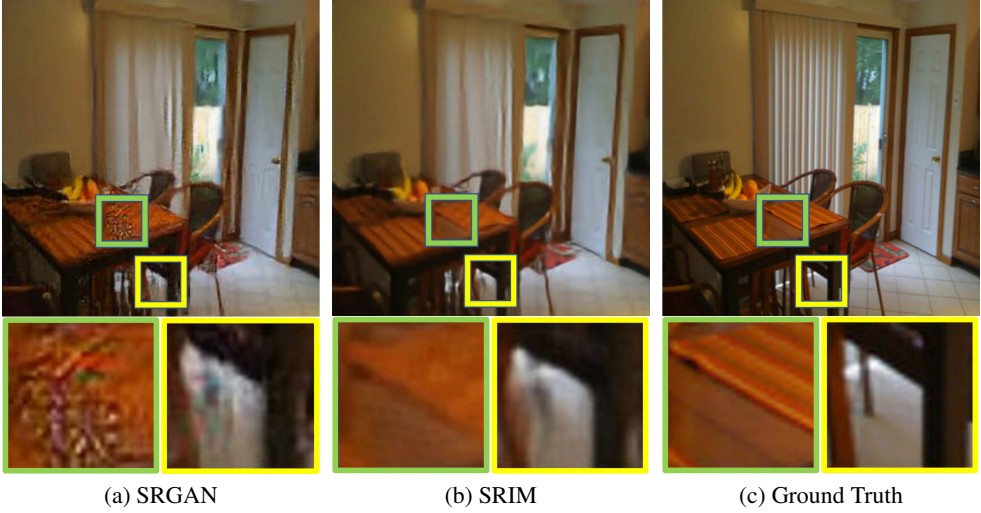

(a) SRGAN          (b) SRIM          (c) Ground Truth

Figure 5: Qualitative comparison of the results of (a) SRGAN, (b) SRIM and (c) the ground truth. High-frequency noise and spurious colours are visible on the table surface and near the chair leg respectively in the output of SRGAN, whereas they are not present in the output of SRIM.

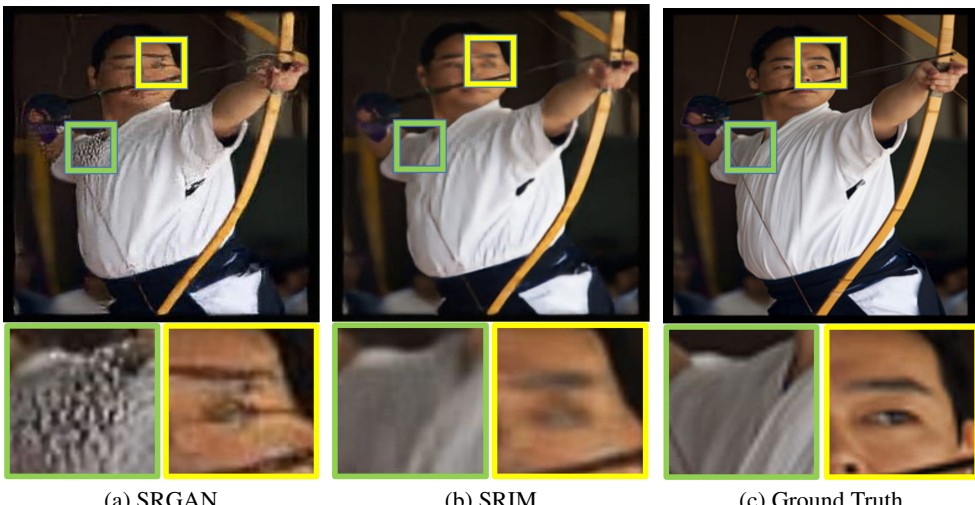

|            | (a) SRGAN | (b) SRIM | (c) Ground Truth |

Figure 6: Qualitative comparison of the results of (a) SRGAN, (b) SRIM and (c) the ground truth. High-frequency noise is visible on the shirt and the shapes of the eye and the eye brow are distorted in the output of SRGAN, whereas these issues are not present in the output of SRIM.

| Data Source | Test 1 | | Test 2 | | Test 3 | |
|---|---|---|---|---|---|---|
| | SRIM | SRGAN | SRGAN | Bicubic | SRIM | Bicubic |
| Seen Classes | **0.63** ± 0.34 | 0.37 ± 0.34 | **0.53** ± 0.36 | 0.47 ± 0.36 | **0.70** ± 0.35 | 0.30 ± 0.35 |
| Unseen Classes | **0.66** ± 0.32 | 0.34 ± 0.32 | **0.53** ± 0.37 | 0.47 ± 0.37 | **0.65** ± 0.34 | 0.35 ± 0.34 |
| All | **0.65** ± 0.33 | 0.35 ± 0.33 | **0.53** ± 0.36 | 0.47 ± 0.36 | **0.67** ± 0.34 | 0.33 ± 0.34 |

Table 2: Results of human evaluation. Three pairwise tests were performed; for each test, super-resolved images using two different methods were presented side-by-side and participants were asked to select the image that seemed more realistic. The mean and the standard deviation of the proportion of images generated by each particular method that participants found to be more realistic is shown, where the mean and the standard deviation were computed over the different participants.

We also run an experiment where we generate multiple samples for the same input image. We can see in Figure 7 that our model (SRIM) produces a greater variety of results compared to SRGAN. In the results produced by SRIM, for regions that are blurrier/less informative in the input image, like the area near the edge of the butterfly wing, the pattern of the white dots is changing across different samples. On the other hand, for regions that are less ambiguous, like the central area of the wing containing thick black lines, the details do not change across different samples. This suggests that the SRIM model is able to learn the different modes of the conditional distribution, while avoiding modelling spurious modes.

### 3.2.4 HUMAN EVALUATION

We conducted a study on 31 human subjects to gauge their opinions on the realism of the outputs of each method. The test consists of three pairwise comparisons: the first test is between our method and SRGAN, the second test is between SRGAN and bicubic interpolation, and the last test is between our method and bicubic interpolation. For each test, we presented participants with 20 pairs of images, with one image in the pair generated by one method, and the other generated by the other method. For each pair, we asked them to choose the image that looks more realistic. Each pair of images is from a different class; 10 are from classes that are present in the training data, and the other 10 are from classes that are unseen. The images in each pair are presented in a random order and the images from the classes in the training set are mixed with those from unseen classes. In Table 2, we show the proportion of outputs of each particular method that are chosen, averaged over different participants. We also break down the results for images that are in/not in the classes seen

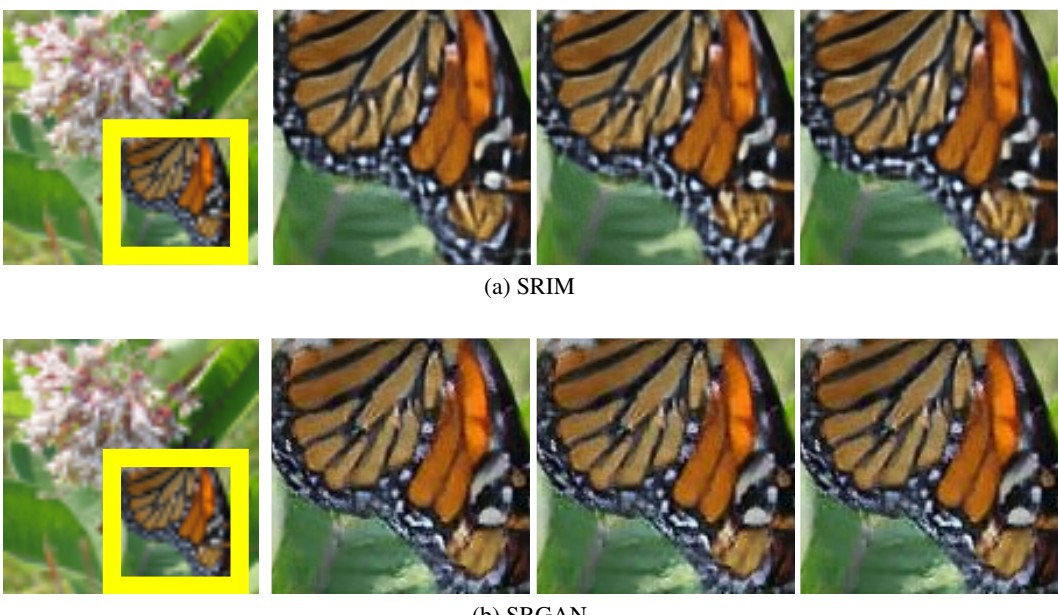

(a) SRIM

(b) SRGAN

Figure 7: Different samples for the same input image for (a) SRIM and (b) SRGAN. The pattern of white dots at the edge of the wing is different for different samples of SRIM, whereas it is identical in different samples of SRGAN. At the same time, the the different samples of SRIM are all plausible and are consistent with the input image.

during training. Because of the limited number of participants and the subjectivity of the criterion of realism, the standard deviation across all tests is relatively high.

We can see from the results of Test 1 that the mean proportion of SRIM outputs being chosen over SRGAN outputs is noticeably higher, which suggests that SRIM is able to produce more realistic results. In addition, on both the classes seen during training and those that are new, SRIM performs similarly, which suggests that SRIM did not overfit to the classes seen during training. As shown by Test 2, only about half of the SRGAN outputs are perceived to be more realistic. This indicates that even though the SRGAN outputs contain sharper edges than bicubic interpolation outputs, the various artifacts lead to a diminished perception of realism. Test 3 results show that SRIM results are perceived to be more realistic than bicubic interpolation results, which demonstrates that our model (SRIM) is able to improve the perceptual quality of the image without introducing artifacts that would cause a degradation in the perception of realism.

## 4 CONCLUSION

In this paper, we proposed a new method for super-resolution, known as Super-Resolution Implicit Model (SRIM), based on the recent proposed method of Implicit Maximum Likelihood Estimation (IMLE). We showed that our method is able to avoid common artifacts produced by existing methods, such as high-frequency noise, colour hallucination and shape distortion. We demonstrated that SRIM is able to outperform SRGAN in terms of standard metrics, PSNR and SSIM (Wang et al., 2004), and human evaluation of realism on the ImageNet dataset. Moreover, we found that SRIM is able to generate multiple plausible hypotheses on the ambiguous regions of the input, while being able to avoid modelling spurious modes.

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

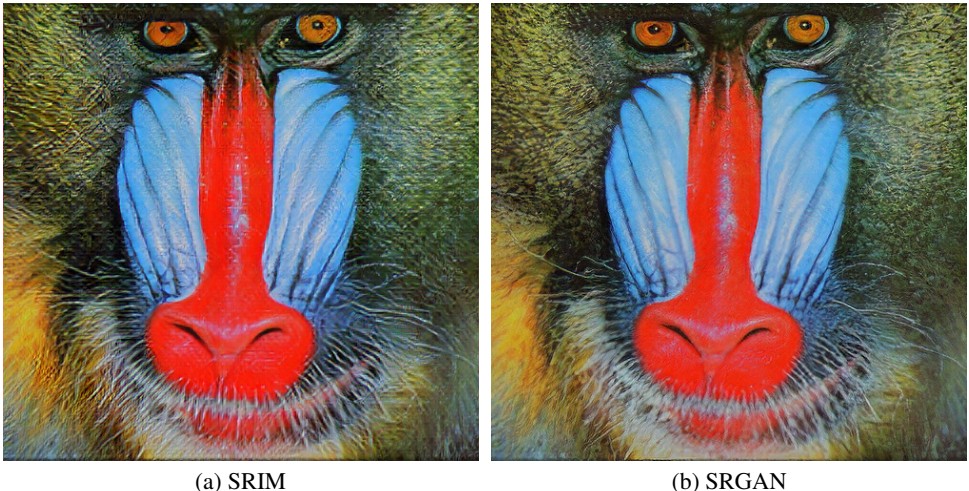

(a) SRIM    (b) SRGAN

Figure 8: Comparison of SRIM and SRGAN under the high-resolution setting of Set14. The input resolution is $120 \times 125$, which is almost $4\times$ the number of pixels in the original setting considered in the main paper, which used input images of size $64 \times 64$. Because this setting is easier, all methods perform better under this setting than under the original setting and so it is more difficult to discern the relative advantages of different methods. As shown, the proposed method and SRGAN perform comparably under this setting.

## APPENDIX A    HIGHER-RESOLUTION INPUTS

In our paper, we focused on the setting that takes a relatively low-resolution ($64 \times 64$) image as input, because it is a more challenging setting that makes the advantages and shortcomings of various methods more apparent. In response to the reviews, we also evaluate our method on Set14, which uses higher-resolution images as input. This is an easier setting because a lot more pixels are provided as input. Under this setting, all methods perform better than under the original setting considered in the main paper and so artifacts are less perceptible, which makes relative performance comparisons more difficult. In Figure 8, we show the results of the proposed method and SRGAN on Set14 and find that they perform comparably.

## APPENDIX B    COMPARISON TO OTHER BASELINES

In response to the reviews, we also compare SRIM and to several other recent methods, including EnhanceNet (Sajjadi et al., 2017), SFT network (Wang et al., 2018a), EDSR network (Lim et al., 2017) and RDN (Zhang et al., 2018). Results are shown in Figure 9.

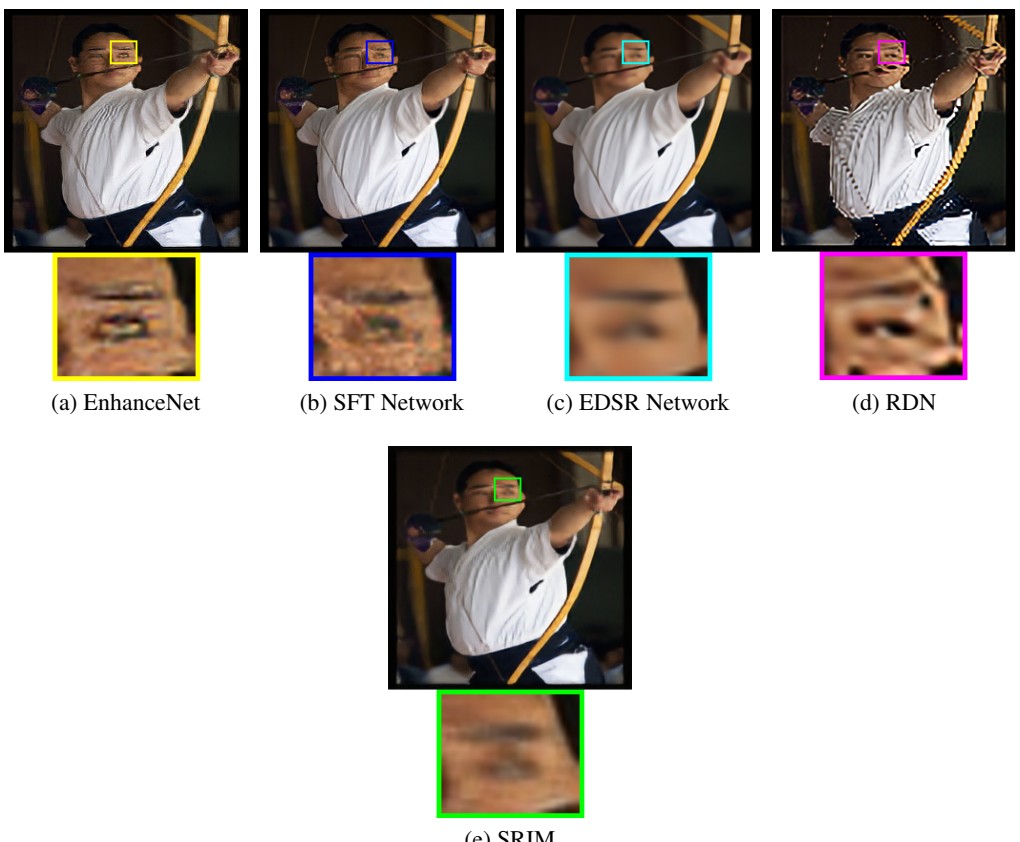

Figure 9: Comparison of SRIM to EnhanceNet, SFT network, EDSR network and RDN. There is shape distortion in the results of EnhanceNet and RDN, high-frequency noise in the results of EnhanceNet and SFT network and blurred details in the result of EDSR network. Note that RDN super-resolves by a factor of 3 rather than 4, and so a higher-resolution input is provided to RDN that results in the same output resolution.

