# OpenReview forum: "Super-Resolution via Conditional Implicit Maximum Likelihood Estimation"
_ICLR.cc/2019/Conference_

### Official Review · AnonReviewer3 · 2018-10-20
**A super-resolution method with with fewer visual artifacts than the SRGAN method.**

**Rating:** 6
**Confidence:** 5

**Review:**

Quality: The overall quality of this paper is good. It adopts a simple but novel idea on SISR and shows clear improvement against existing method (e.g., SRGAN).

Clarify: This paper is well written and easy to follow. It shows a clear motivation for adopting the implicit probabilistic model.

Originality: To the best of my knowledge, this paper is the first work to learn multi-modal probabilistic model for SISR.

Significance: While the results can be further improved (still look a bit blurred), this paper shows an interesting and important direction to learn better mappings for SISR.

Pros:
+ The writing is clear.
+ The proposed method is well motivated and easy to understand.
+ The experimental results include both objective and subjective evaluations.

Cons:
- The two-stage architecture is similar to the following generative models and SR methods. It’s suggested to discuss them as well.
[a] Denton, E. L., Chintala, S., & Fergus, R. “Deep generative image models using a￼ laplacian pyramid of adversarial networks”. NIPS, 2015.
[b] Karras, T., Aila, T., Laine, S., & Lehtinen, J. “Progressive growing of gans for improved quality, stability, and variation”. ICLR 2018.
[c] Lai, W. S., Huang, J. B., Ahuja, N., & Yang, M. H. “Deep laplacian pyramid networks for fast and accurate super-resolution.” CVPR 2017.
[d] Wang, Y., Perazzi, F., McWilliams, B., Sorkine-Hornung, A., Sorkine-Hornung, O., & Schroers, C. “A Fully Progressive Approach to Single-Image Super-Resolution.”. CVPR Workshops 2018.

- In the hierarchical sampling (section 2.4), it’s not clear how to generate the upper noise vector “conditioned on the lower noise vector”.

- The hierarchical sampling seems to improve the efficiency of training. I wonder does it affect the results of testing?

- In the implementation details (section 2.5), I don’t understand why you need to transfer the the feature activations from GPU to CPU? I think all the computation can be done on GPU for most common toolboxes. Projecting the activations to a lower dimension with a “random Gaussian matrix” sounds harmful to the results.

- How do you generate the low-resolution images? Are you using bicubic downsampling or other approaches? This detail should be clarified.

- While the evaluation with PSNR and SSIM is a reference to show the quality, many literatures already show that PSNR and SSIM are not correlated well with human perception. It is suggested to also evaluate with some perceptual metrics, e.g., LPIPS [e].
[e] Zhang, R., Isola, P., Efros, A. A., Shechtman, E., & Wang, O. “The unreasonable effectiveness of deep features as a perceptual metric.” CVPR 2018.

- In Figure 7, how do you generate different results from the same input image for SRGAN? From my understanding, SRGAN doesn’t take any noise vector as input and cannot generate multi-modal results.

- I feel that the comparison with only SRGAN is not enough. There are some GAN-based SR methods [f][g]. It’s also suggested to compare with MSE-based state-of-the-art SR algorithms [h][i].

[f] Sajjadi, M. S., Schölkopf, B., & Hirsch, M. “Enhancenet: Single image super-resolution through automated texture synthesis.“ ICCV 2017.
[g] Wang, X., Yu, K., Dong, C., & Loy, C. C. “Recovering realistic texture in image super-resolution by deep spatial feature transform.” CVPR 2018.
[h] Lim, B., Son, S., Kim, H., Nah, S., & Lee, K. M. “Enhanced deep residual networks for single image super-resolution.” CVPR Workshops 2017.
[i] Zhang, Y., Tian, Y., Kong, Y., Zhong, B., & Fu, Y. “Residual dense network for image super-resolution.” CVPR 2018.

---

> ### Author Response · Authors · 2018-11-28
> **Response to your review**
>
> Thank you for your review. We have updated our paper to include a discussion of other methods that use a multi-stage architecture in section 2.3. To generate the upper noise vector, we generated samples with the noise input to the lower sub-network fixed and then perform kNN search among this pool of samples. Using hierarchical sampling during training improves the results at test time compared to vanilla sampling using the same number of samples, because it functions as if more samples were generated using vanilla sampling at training time, which in the context of IMLE results in improved performance. We tried performing all computation on the GPU, but ran into memory issues with our implementation (which may be an issue specific to our implementation). It’s possible that projection using a random Gaussian matrix could introduce distortions, but we also tried using the original features without projection and observed no significant difference in the results (which can also be explained theoretically by the Johnson-Lindenstrauss lemma). The low-resolution input is generated by applying Gaussian blur and subsampling. In our experience, LPIPS focuses more on the high-level semantics and less on the low-level details, and so does not correlate well with human judgement in the context of super-resolution. We conducted human evaluation to eliminate possible biases that can be introduced by the choice of evaluation metric. To generate different samples from the same input image for SRGAN, we trained a SRGAN model where we add a second input to the generator containing random noise. We found that even when with random noise as input, SRGAN cannot generate multi-modal results due to mode collapse. As per your suggestion, we have updated the paper to include comparisons to [f][g][h][i] in the appendix and found that our method outperforms these methods in terms of image quality.

---

### Official Review · AnonReviewer1 · 2018-10-29
**Paper is written well and except for some sections, it provides enough details. The work is original enough but might need some improvement or more explanation in experiments/result section.**

**Rating:** 6
**Confidence:** 5

**Review:**

This paper proposes a technique to find a maximum-likelihood estimate of the super-resolved images under latent variables without computing it. Paper is mostly clearly written and except for some sections, it provides enough details. The work is original enough but might need some improvement or more explanation in experiments/result section.

Pros:
-The idea seems to be original enough, simple and easy to implement.
-A nice follow-up of the recent work in NN search and Implicit maximum likelihood estimation.
-Many details that could be helpful for further research in the area are given.

Cons:
-Regarding methodology, an unclear point in the paper is how different networks trained according to algorithm 1. Is each sub-network trained separately? Is the visual perception based feature space pre-trained and fixed, or is it jointly retrained with the super-resolution network?

-Another critical point is post-training, particularly the way learned parameters are used could be explained better: Given a super-resolution model f, how the super-resolution of a single image is performed? What is the sampling variation? How likely such a network can be productionized in real-time systems (e.g., digital displays or embedded systems)? How does the proposed approach compared to GAN based methods with regards to that? Is multi-modality a problem in this case? Any way to choose one specific mode in a conscious way?

-My main concern about the paper is the results section: Authors perform both large-scale offline comparison (imagenet) and a small subset human evaluation. The results in human evaluation need some explanation. This comparison is identical to several previous 1-1 comparisons performed in literature and almost every single such comparison it has been found that state of the art techniques (e.g., 10+ years of super-resolution algorithms) significantly outperform bicubic interpolation. However, Table 2 in the paper suggests that both SRGAN and SRIM barely beats bicubic interpolation. For example, authors in https://arxiv.org/pdf/1209.5019.pdf showed that a relatively older supervised technique beat bicubic 90% of the time. There seems to be some explanation needed here: Is it the sample size? Are the samples from both SRIM and SRGAN very variable?

---

> ### Author Response · Authors · 2018-11-28
> **Response to your review**
>
> Thank you for your review. To train the sub-networks, we first train the sub-network for the lower input resolution; then we add the second sub-network on top and train both sub-networks jointly. The feature space is pre-trained and fixed in our setting.
>
> At test time, given a low-resolution input image, we randomly generate one noise input for each sub-network and feed all inputs into the network to get the super-resolved output. The variation exhibited by different samples corresponds to different plausible ways to super-resolve ambiguous regions of the input image. To productionize such a network in real-time systems, we can use established techniques for model compression and binarization. Our approach is as easy to productionize as GAN-based methods because the sampling procedure for our method and GAN are the same at test time (only the training algorithm differs). Multi-modality could be a problem in this case because it is important in various applications to have control over which mode we want to output; for example, if we’d like to super-resolve all frames in a video, we need to make sure the same mode is selected consistently across all frames, so that the blurry regions are super-resolved in the same way in all frames. To choose one specific mode in a conscious way, we can simply choose a noise input that results in a super-resolved image that we prefer and use this (fixed) noise input for all low-resolution input images.
>
> In our paper, we focused on a more challenging setting than that is typically considered in the literature, where the input image is of a relatively low resolution (64x64). We chose this setting because most existing methods already perform very well when the input is of a higher resolution, and so the differences between different methods are easier to discern under a more challenging lower-resolution setting. Because the limitations of SRGAN are more perceptible under this setting, SRGAN does not outperform bicubic interpolation by a large margin; however, SRIM does outperform it by a fairly large margin.

---

### Official Review · AnonReviewer2 · 2018-11-02
**Interesting approach, but experiments are not appropriate**

**Rating:** 5
**Confidence:** 3

**Review:**

- Summary
This paper proposes a method based on implicit maximum likelihood estimation for single-image super-resolution. The proposed method aims at avoiding common artifacts such as high-frequency noise and shape distortion. The proposed method shows better performance than SRGAN in terms of PSNR, SSIM, and human evaluation of realism on the ImageNet dataset.

- Pros
  - The proposed method shows better performance than SRGAN in terms of PSNR, SSIM, and human evaluation.
  - The selection of the evaluation methods is appropriate. In the field of image super-resolution tasks, both signal accuracy (e.g., PSNR) and perceptual quality (e.g., human evaluation) are important.

- Cons
  - The experiments are conducted thoroughly in the ImageNet, but the selection of the dataset is not appropriate. It would be better to apply the proposed method to other datasets which are used recent papers.
  - Also, the selection of the methods to be compared is not appropriate. It would be better to provide recent state-of-the-art methods and compare the proposed method with them.

The proposed approach is interesting and promising, but the selection of the methods and datasets to be compared is not appropriate.

---

> ### Author Response · Authors · 2018-11-28
> **Response to your review**
>
> Thank you for your review. We note that the existing datasets commonly used in the super-resolution literature for evaluation are quite small (they typically contain <=100 images), whereas ImageNet is a lot bigger and can provide more reliable results. Common super-resolution testing datasets are also typically used in a high-resolution setting (i.e. fairly high-resolution inputs are fed into the super-resolution algorithm), whereas our experiments are conducted with 64x64 inputs, which can contain 3-6x fewer pixels than the inputs that are used with common datasets. Because the typical inputs are at higher resolutions, the typical setting is easier and so differences between different methods are harder to discern, which is why we performed evaluation under a more challenging setting. However, as per your request, we have updated our paper to include a comparison of our method to SRGAN on the commonly used dataset of Set14 in the appendix and found that both methods perform comparably, but note that this comparison is less informative because all methods perform well on Set14. We have also updated our paper to include comparisons to more recent methods in the appendix, including EnhanceNet, SFT network, EDSR network and RDN. We found that our method outperforms these methods in terms of image quality.

---

### Public Comment · (anonymous) · 2018-10-04
**Relation to prior work**

Interesting paper. Please note that there has been significant progress in this field since SRGAN, see for example https://arxiv.org/pdf/1809.07517.pdf

---

> ### Author Response · Authors · 2018-10-09
> **Response to your comment**
>
> Thanks for your comment. We note that the referenced paper reports on the performance of methods submitted to a recently concluded ECCV workshop challenge. Because the methods were only released a week before the submission deadline, whose code and implementation details remain unavailable in many cases, we weren’t able to compare to these methods. We do note that Figure 2 in the referenced paper shows that SRGAN is one of the best available methods in terms of visual quality at the time the challenge was conducted.

---

> > ### Public Comment · (anonymous) · 2018-10-09
> > **Clarifying the comment**
> >
> > Thank you for your response. I was referring to the overview of the field in the first pages of the referenced paper. Some prior works mentioned include EnhanceNet - Sajjadi et al. ICCV 2017, SFTGAN - Wang et al. CVPR 2018, ProGAN - Wang et al. CVPR Workshops 2018. These three have released their models publicly.

---

> > > ### Author Response · Authors · 2018-10-12
> > > **Response to your clarification**
> > >
> > > Thanks for your clarification. EnhanceNet is similar to SRGAN in terms of perceptual quality but has higher reconstruction error, as shown by Figure 2 in the referenced paper. SFTGAN uses a semantic segmentation model to predict the categories of objects and therefore uses auxiliary supervision in the form of segmentation masks. Our method does not use such supervision, and so cannot be directly compared. ProGAN is somewhat lesser known; thanks for bringing it to our attention. We ran their model and found that the results exhibit similar types of artifacts as those of SRGAN.

---

### Public Comment · (anonymous) · 2018-10-07
**Non-standard evaluation dataset**

I enjoyed reading your paper, your approach is quite novel in my opinion. One question: why is the dataset used for evaluation in your paper not those commonly used for testing super resolution algorithms? The commonly used datasets for image super-resolution are BSD100, Urban100 and DIV2K, as in SRGAN (Ledig et al. CVPR17), EDSR (Lim et al. CVPRW17), EnhanceNet (Sajjadi et al. ICCV17), DBPN (Haris et al. CVPR18), etc. These datasets allow a straightforward comparison with others.

Also, there is no official implementation of SRGAN, but there are officially published results of SRGAN on the BSD100 dataset (there is a link in the SRGAN paper). Evaluating on the BSD100 dataset would allow a comparison with the original SRGAN algorithm, and not with a reproduction attempt from Github.

---

> ### Author Response · Authors · 2018-10-09
> **Response to your comment**
>
> Thanks for your comment. As is standard in machine learning, the training and test sets are from the same collection of images (ImageNet) to eliminate any possibility of biases in the evaluation results due to domain shift. We chose to use ImageNet for training because that was used by SRGAN, and this choice mandated the use of ImageNet for testing.
>
> Nevertheless, as per your suggestion, we evaluated our method on BSD100 and found the results were comparable to those on ImageNet: SSIM was 0.7254 (compared to 0.7153 on ImageNet) and PSNR was 26.39 (compared to 25.36 on ImageNet).

---

### Comment · Area_Chair1 · 2018-12-10
**Baseline comparison**

The main novelty of the paper lies in using multiple noise vectors to reconstruct the high resolution image in multiple ways. Then you select the reconstruction with minimal loss and update the parameters to improve the fit for the best reconstruction. I think this is a neat idea, but for completeness, an important control experiment is using the same architecture with only m=1 noise vector (i.e., using a constant noise vector all the time). Have you demonstrated the benefit of your approach over this baseline?

--AC

---

> ### Author Response · Authors · 2018-12-10
> **Yes, using multiple noise vectors is important**
>
> Yes, we have tried using m=1, but found that this resulted in blurrier images because not allowing the net to output multiple possibilities essentially forces it to predict the mean of the different possibilities. We'll include this result in the camera-ready.

---

### Meta-Review · Area_Chair1 · 2018-12-14
**The paper needs to be improved**

**Confidence:** 4
**Recommendation:** Reject

**Metareview:**

The main novelty of the paper lies in using multiple noise vectors to reconstruct the high resolution image in multiple ways. Then, the reconstruction with minimal loss is selected and updated to improve the fit against the target image. The most important control experiment in my opinion should compare this approach against the same architecture with only with m=1 noise vector (i.e., using a constant noise vector all the time). Unfortunately, the paper does not include such a comparison, which means the main hypothesis of the paper is not tested. Please include this experiment in the revised version of the paper.

PS: There is another high level concern regarding the use of PSNR or SSIM for evaluation of super-resolution methods. As shown by "Pixel recursive super resolution (Dahl et al.)" and others, PSNR and SSIM metrics are only relevant in the low magnification regime, in which techniques based on MSE (mean squared error) are very competitive. Maybe you need to consider large magnification regime in which GAN and normalized flow-based models are more relevant.